# Estimation of Airflow Parameters for Tail-Sitter UAV through a 5-Hole Probe Based on an ANN

**DOI:** 10.3390/s23010417

**Published:** 2022-12-30

**Authors:** Xiaoda Li, Yongliang Wu, Xiaowen Shan, Haofan Zhang, Yang Chen

**Affiliations:** 1Department of Mechanics and Aerospace Engineering, College of Engineering, Southern University of Science and Technology, Shenzhen 518055, China; 2School of Aeronautics and Astronautics, Xihua University, Chengdu 610039, China; 3College of Innovation and Entrepreneurship, Southern University of Science and Technology, Shenzhen 518055, China; 4School of Physics and Mechatronics Engineering, Longyan University, Longyan 364012, China

**Keywords:** tail-sitter VTOL, estimation of airflow parameters, large angle of attack, 5-hole probe, neural network

## Abstract

Fixed-wing vertical take-off and landing (VTOL) UAVs have received more and more attention in recent years, because they have the advantages of both fixed-wing UAVs and rotary-wing UAVs. To meet its large flight envelope, the VTOL UAV needs accurate measurement of airflow parameters, including angle of attack, sideslip angle and speed of incoming flow, in a larger range of angle of attack. However, the traditional devices for the measurement of airflow parameters are unsuitable for large-angle measurement. In addition, their performance is unsatisfactory when the UAV is at low speed. Therefore, for tail-sitter VTOL UAVs, we used a 5-hole pressure probe to measure the pressure of these holes and transformed the pressure data into the airflow parameters required in the flight process using an artificial neural network (ANN) method. Through a series of comparative experiments, we achieved a high-performance neural network. Through the processing and analysis of wind-tunnel-experiment data, we verified the feasibility of the method proposed in this paper, which can make more accurate estimates of airflow parameters within a certain range.

## 1. Introduction

The tail-sitter aircraft can cruise at a relatively high speed like a fixed wing aircraft, and it can also take off and land vertically like a rotary wing aircraft, so it has attracted much attention in recent years. However, it is precisely this advantage that makes it have to go through the phase of attitude change in the flight process, which makes the tail-sitter aircraft more vulnerable to the wind and more difficult to control. Therefore, a good ability to measure wind disturbance is needed for tail-sitter aircraft. In fact, the measurement of wind disturbance is essentially the measurement of wind speed, which can be converted into the measurement of air speed and ground speed, and then into the measurement of air flow parameters (including angle of attack, sideslip angle and incoming flow speed), because the ground speed can be simply obtained through GPS [1]. For the traditional fixed wing aircraft, the measurement of these parameters is relatively simple, but the traditional measurement devices, such as a pitot tube and vane angle sensor, are not completely applicable to the tail-sitter aircraft, as they are unsuitable for large-angle measurement [2,3]. In addition, their performance is poor when the UAV flies at a low speed [2]. (For VTOL aircraft, such as rotor-wing or fixed-wing VTOL in hovering or transition mode, the speed is normally below 25 m/s (90 km/h). This speed is usually less than that of the fixed-wing aircraft or fixed-wing VTOL aircraft in cruise mode. Thus, here in this article, what we talk about is the situation of the speed being below 25 m/s, which can be regarded as low speed, and the speed being above 25 m/s, which can be regarded as high speed.) Therefore, in the field of parameter measurement for small UAVs, people more often choose to use a multi-hole probe to replace traditional measurement equipment [4].

Similarly, we chose a 5-hole probe to estimate the airflow parameters. Using the pressure data of the five holes at the probe’s head measured by pressure sensors, the airflow parameters can be estimated by many different methods, including the look-up table method, interpolation method, etc, which are briefly introduced below.

The table lookup (figure) method and interpolation method are similar [5,6,7,8,9,10]. They obtain pressure data and corresponding air flow parameters through a large number of experiments, and then establish the database (can be polynomial, table and image). Then, using the database, the flow parameters can be obtained from the pressure data. They are simple in principle, but more advanced experiments are needed, and they usually perform well when the angle to be estimated is less than 45°.

The algebraic solution is usually simple. The least square method and three-point method [11,12,13,14,15,16] establish the aerodynamic model of the probe surface, obtain the key parameters through the experiment and then transform the prediction problem into a simple problem that can be solved algebraically. However, in order to improve the accuracy of these models, they not only need more experiments, but also need to establish the relevant error model. In addition, there will be an iterative process in the process of solving. They are more suitable for small angles (less than 30°).

The method of polynomial fitting [17] is generally to use third-order, fourth-order or higher-order polynomials to convert the experimental data into dimensionless parameters for fitting. It can be said to be the simplest method for practical applications. However, because of the oversimplification, it only works for small angles (less than 25°).

Artificial neural networks have been very popular in recent years. Considering the obvious relationship between the parameters we need to estimate and the pressure data, the neural network method can be regarded as a good method. It is also true that we can get well-performing neural networks only through structural design, data collection and network training. In addition, of all methods, neural network methods have the smaller error, fastest calculation speed and lowest memory needs. However, previous studies focused more on the performances of neural networks at small angles (most of them below 25°, but a few of them have reached 50°) [18,19,20,21,22,23,24,25].

Our aircraft of interest, tail-sitter aircraft, need to change their mode during flight to achieve the purpose of combining the advantages of fixed-wing aircraft and rotary-wing aircraft, which leads to the need for accurate prediction of airflow parameters when the angle of attack is large. Therefore, in this paper, we explore the performance of the five-hole probe based on the neural network method at large angles.

In this article, our contributions are as follows: we propose a combined airflow-parameters-estimation method using a 5-hole probe and an ANN estimation algorithm to offer a solution to face the challenge of large AOAs (angles of attack) and AOSs (angles of sideslip) or estimation for the tail-sitter UAV. As is known, there is almost no prior work on large AOA and AOS estimation for tail-sitter UAV. Analysis results show that the proposed method has good performance: the error of the angle of attack can be found within ±1° in the range of −60° to 60°; the sideslip-angle-estimation error is larger but can be less than ±1° when the absolute value of sideslip angle is less than 50°; and the estimation error of incoming flow velocity can be kept relatively small in the whole measurement range. This article is an extension of the work presented at the AIAA Aviation Forum [26], but article [26] only discusses the sideslip-angle estimation when the angle of attack is zero and the angle-of-attack estimation when the sideslip angle is zero. In this paper, we have used more experimental data; optimized the data itself and the structure of the neural network used in data processing; and further explored the situation where the angle of attack and sideslip angle are not zero at the same time (that is, the aircraft is facing flow from all directions).

In this paper, the research purpose, background and results are presented first. Section 2 describes the probe’s structure and the estimation method based on ANN, which is followed by the design of the wind-tunnel experiment. The optimization of estimation method based on ANN is introduced in detail in Section 4. Section 5 is about the analysis of experimental data, and the conclusion of our work is in last section.

## 2. The Tail-Sitter VTOL, 5-Hole Probe and ANN-Based Estimation Method

### 2.1. The Tail-Sitter VTOL

Here is the research background for our article. This tail-sitter fixed-wing VTOL UAV, “ETS-20,” was developed at the Southern University of Science and Technology (SUSTech), China, by the research group of Professor Xiaowen Shan, as shown in Figure 1. The ETS-20 tail-sitter VTOL UAV is characterized by long endurance (more than 4 h), a large payload allowance (more than 1.5 kg), a long range (more than 240 km), a good tail-vector stability control effect, a simple structure and easy disassembly. It is particularly suitable for marine monitoring, forest fire prevention, emergency rescue, long-term air surveillance and other purposes.

Compared with the traditional fixed-wing aircraft, the combination of a propeller and control surface makes it possible to control the flight attitude at full airspeed, thereby ensuring its maneuverability and vertical takeoff and landing capability. However, at the same time, this structure also makes it very sensitive to the propeller speed, wind speed and its own angle of attack, so it is very important to estimate the airflow parameters.

### 2.2. The 5-Hole Probe

We used a 5-hole pressure probe method based on an ANN to estimate the airflow parameters the tail-sitter VTOL vehicles require. The probe was the source of measurement data, and the pressure of the incoming flow was sensed by the pressure sensor through the holes on the head of probe for subsequent calculations.

In consideration of cost, we designed the probe we used (Figure 2) in the experiment by referring to the existing five-hole probe (manufactured by VECTOFLOW GMBH in Germany), and then made it by 3D metal printing. The five pressure holes at the top can obtain pressure data at five points at fixed positions on the semi-circular sphere (except the pressure hole at the top, the pressure holes are evenly distributed, and these four holes have a cone angle of 55°). Meanwhile, static pressure information can be obtained through the static pressure holes on the side.

By comparing the pressure difference between each pressure orifice and the static pressure orifice, we can obtain a set of pressure data characterizing the air flow speed and direction. By matching the multiple groups of pressure data obtained in the measurement process with the actual angles recorded in the experiment, we can obtain the relationship model between the pressure difference and the velocity and direction of the airflow, so that we can predict the airflow parameters according to the pressure data later.

### 2.3. The ANN-Based Estimation Method

Our main task was to establish the relationship model between the pressure values and the air flow parameters. As mentioned in the introduction, there are many methods to build the model we need, such as the look-up table, interpolation, polynomial fitting and neural network methods. After comparison, considering the accuracy and scope of application, we finally chose the neural network method to establish the estimation model.

Among various neural networks, we chose the BP neural network as the basis. BP is backpropagation. The backpropagation algorithm is a parameter-training method suitable for multi-layer neural networks, and its basis is gradient descent. In the actual training process, we cannot directly obtain the errors of the nodes of the hidden layer, but only the errors of the prediction results and labels are directly known. Therefore, the known output errors need to be transformed into the errors of the hidden layer through layer propagation, and various weight parameters of the network should be adjusted in this process, so as to realize the network training.

Figure 3 is a simplified diagram of a BP neural network structure. It can be seen that a BP neural network consists of an input layer, an output layer and a hidden layer. External information, that is, experimental data, enters the neural network through the input layer and is transmitted to the hidden layer for processing. There is no limit on the number of hidden layers, which can be single layer or multi-layer. It can process the input data through predetermined parameters, so as to extract the features in the data. The output layer at the end, after a specific activation function, outputs the predicted value of the entire neural network. The so-called forward propagation is the process of a group of data passing through the input layer, reaching the hidden layer to be processed and output through the output layer.

However, generally speaking, the first forward-propagation results always have a big deviation, so the above-mentioned, namely, the most important backpropagation process in a BP neural network, is needed. Through backpropagation, the parameters and weights in each layer are corrected, making the errors gradually decrease via gradient descent. The training process of BP neural network is the process of error-based repeated propagation and weighted repeated adjustment, which finally makes the output meet the requirement or the training time meet the requirement.

However, the specific structure of the neural network still needs to be considered in detail. An excellent neural network architecture can help us get more accurate estimation results. Additionally, for a neural network, the main factors affecting its structure include: the choice of input and output, the number of hidden layers and the number of neurons in each layer. Therefore, the exploration of the neural network structure we want is based on these three parts.

## 3. The Wind-Tunnel Experiment

The artificial neural network method needs abundant data for training. We conducted the wind-tunnel experiment to obtain the required experimental data.

The multi-hole pressure probe used in the experiment was made by 3D metal printing technology and fixed to the turntable using tooling (Figure 4).

In order to realize the change in a large angle range, we used a two degree of freedom turntable, whose motors in both directions can be controlled by a computer (Figure 5).

We used the VectoDAQ Air Date Computer (Figure 6) to obtain the pressure data, whose measurement error is less than 0.5%. It can be connected to a computer through a USB and transfer the collected data to the software that processes the data.

In the experiment, the turbulence of the wind tunnel we used was less than 0.1%, which can provide a stable wind-field environment.

As for the speed setting in the experiment, first of all, we mainly focused on the performance of the research object being at low speed (below 25 m/s). Second, we need to consider how it behaves over the whole low-speed range. Finally, we found in the pre-experiment that when the speed was too small (such as 5 m/s, which is not shown), the data availability was poor (the pressure fluctuation of a single measuring point was similar to the pressure change between adjacent measuring points), so 10, 15 and 20 m/s were finally selected as the velocities in the experiment.

As for the selection of the experimental angle range, the tail-sitter VTOL must undergo attitude change during flight, which may cause it to encounter an angle of attack close to 90°. Thus, we want to get the same performance for the neural network approach at angles of attack ranging from 50° to 80° and even up to 90°. Therefore, the initial experimental plan was to set the variation range for both the angle of attack and the sideslip angle to −90° to 90°. However, due to the limitation of the wind-tunnel test section (Figure 7), we could only set the range of angle of attack to −50° to 80° and that of sideslip angle to −30° to 30°.

## 4. Optimization of Estimation Method Based ANN

Obviously, different neural networks will show different performances in the process of building models due to their differences in structure. The contents to be considered include the selection of input and output, the number of hidden layers and the number of hidden layer neurons.

The first is the selection of input and output. In one of Quindlen J.’s papers [23], they verified that the accuracy of multi-output artificial neural networks was weaker than that of multiple single-output artificial neural networks through comparative experiments. Therefore, in order to realize the estimation of the angle of attack, angle of sideslip and velocity of the incoming flow, we segmented the data and trained the corresponding neural networks with three different outputs using the corresponding data.

Then, the number of hidden layers and the number of neurons in each layer were selected. At present, there is no mature theory to help determine these parameters, so we could only choose them according to experience, mainly based on two empirical theories. The first is that the higher the number of hidden layers, the better the training effect and the longer the training time; the second is that the distribution of the number of hidden layer neurons has a better training effect if the number of neurons in the hidden layer changes from less to more, and then from more to less.

In order to reduce the time required for training while ensuring the training effect, and also considering that the corresponding relationship between the input and output of our model is relatively complex, we finally chose to set the number of hidden layers to four. For the exploration of the specific number of neurons in each of the four hidden layers, we set several possible values for each layer, then combined them, trained them separately and finally took the combination with the best training performance. The number of optional neurons in the first hidden layer was (5,10,15,20). The second layer had (5,10,15,20,25,30), the third layer had (5,10,15,20,25) and the fourth layer had (5,10). There were 180 combinations in total.

The metrics that can be used in machine learning algorithms include MSE, RMSE, MAE and $R^{2}$. MSE is the mean square error, RMSE is the root mean square error, MAE is the mean absolute error and $R^{2}$ is the coefficient of determination of the prediction. They are defined as
$$MSE=\frac{1}{n}\sum_{i=1}^{n}{\left(y_{predict}-y_{test}\right)}^{2}$$
$$RMSE=\sqrt{\frac{1}{n}\sum_{i=1}^{n}{\left(y_{predict}-y_{test}\right)}^{2}}$$
$$MAE=\frac{1}{n}\sum_{i=1}^{n}\left|y_{predict}-y_{test}\right|$$
$$R^{2}=1-\frac{MSE(y_{predict},y_{test})}{Var(y_{test})}$$

The smaller the values of MSE, RMSE and MAE, the better the quality of the neural network structure, and their minimum value is 0. The larger the value of $R^{2}$, the better the quality of the neural network, and its maximum value is 1. MSE, RMSE, MAE and $R^{2}$ for 180 neural networks with different structures in the paper are shown in Figure 8. The horizontal axis in the figures represents the labels of all combinations, and the vertical axis represents the corresponding scores of different neural networks. The best combination of hidden layer neurons could be obtained by selecting the combination with the highest score from the images, and it is (15,30,15,5).

In addition to the above factors, we note that the quality of input data will also have a certain impact on the training results of neural networks, so we also tested and got the results before and after filtering the input data. The training results show that filtering the input data can improve the training performance obviously. Figure 9 and Figure 10 show a comparative example of training performance before and after filtering the input data.

To sum up, we finally determined that the neural network for the ANN-based estimation method should be a BP neural network with the structure of (5-15-30-15-5-1). Its input is five pressure values, and its output is angle of attack or sideslip angle or speed of incoming flow. There are four hidden layers inside, and the number of neurons in each layer is (15-30-15-5).

We used Neural Net Fitting App in MATLAB R2020b software to build and train the neural network. In order to ensure the training performance and efficiency, we selected the Levenberg–Marquardt algorithm as the training algorithm. After training, the required neural network can be obtained, and it can be imported into Simulink for subsequent verification and simulation. Partial images of this process are shown below (Figure 11 and Figure 12).

## 5. The Analysis of the Experiment Data

### 5.1. The Content of the Experimental Data

In fact, we obtained a lot of data from the wind-tunnel experiments. Here, we look at the experimental data we got when the sideslip angle and the angle of attack were not zero at the same time. The specific measurement contents are described as follows: the angle of attack changed from −50° to 80° at uneven intervals. (Specifically, when the absolute value of the angle of attack is between 20°, a 4° interval was used; when the absolute value of the angle of attack was between 20° and 40°, a 2° interval was used; when the absolute value of the angle of attack was greater than 40°, a 1° interval was used. This is because we were mainly concerned with large angles of attack). At each angle of attack, the sideslip angle changed from −30° to 30° at uniform intervals (2°). For each fixed angle of attack and sideslip angle combination, the pressure measuring valve collected data at a frequency of 20 Hz for 5 s. In addition, we selected three different incoming flow velocities, 10, 15 and 20 m/s.

The purpose of selecting this part of experimental data was to verify that the airflow parameter estimation method based on neural network can be applied to the case of large angles of attack, so as to better meet the needs of tail-sitter VTOL UAV. On the other hand, it is to test whether this method would work well when the angle of attack and the sideslip angle are not zero at the same time (this means that the drone is exposed to non-positive airflow).

### 5.2. The Performance of Filtering

Although the turbulence degree of the wind tunnel is not high, the pressure data obtained from the probe have relatively large fluctuations because of the relatively low incoming flow velocity. In addition, the fluctuation of pressure data gradually intensifies as the angle of attack increases and with the separation of air flow. Theoretically, the pressure data of a certain combination of speed and angle should be fixed, so we can try to use the filtering method to process the data.

The so-called filtering is to simply use the one-dimensional Kalman filter method to process the original data, so as to reduce the fluctuations in the data. We should note that in the training and subsequent application process, the input value should also be the filtered value. Figure 13 is a comparison of the fluctuation of some experimental data before and after filtering, and the smoother curve is the filtered one.

In fact, this simple filtering method will make the experimental data at low angles of attack produce fluctuations that did not exist before, thereby slightly increasing the estimation error. However, in order to improve the overall training performance, we finally chose filtered data for training and testing.

### 5.3. The Performance of Training

After training the neural network with the training set, we can input the test set, compare the obtained output with the expected output and get the following figures.

Figure 14 shows the relationship between the estimation error of angle of attack and the value of angle of attack. From the figure, we can see that as the absolute value of the angle of attack increases, the estimation error gradually increases, and there are relatively good estimation errors within ±60°. When the angle of attack exceeds 70°, the estimation error increases rapidly, but basically remains within ±3° when angle of attack is less than 80°.

Figure 15 shows the relationship between the estimation error of sideslip angle and the value of angle of attack. From the figure, we can see that, as the absolute value of the angle of attack increases, the estimation error gradually increases, and there are relatively good estimation errors within ±50°. When the angle of attack exceeds 60°, the estimation error of sideslip angle will increase rapidly, even exceeding 10°.

Figure 16 shows the relationship between the estimation error of the incoming flow velocity and the value of angle of attack. From the figure, we can see that as the absolute value of the angle of attack increases, the estimation error is basically unchanged and remains at a relatively small level. The reasons for this are that few velocity values were used in the experiment, and more importantly, there is a good and obvious relationship between pressure data and velocity. Therefore, the estimation of the magnitude of the velocity does not require much attention.

In addition, we also obtained the relationship between these errors and the change in sideslip angle (Figure 17, Figure 18 and Figure 19).

It should be noted that the abscissa of the following three images is the change in samples, which represents the uniform change in sideslip angle from −30° to 30° from left to right. We can see that the error displayed in these images always changes from small to large in a certain period. This phenomenon is because in our experimental data, each sideslip angle corresponds to a group of angles of attack varying from −50° to 80°. These images also verify the conclusion that the estimation error of the airflow parameters will increase gradually as the absolute value of the angle of attack increases.

From above figures and analysis, we can draw the following conclusions:(1)When the angle of attack and sideslip angle are not zero at the same time, the neural network method can still well predict the angle of attack, sideslip angle and velocity of incoming flow. It can be seen in Figure 14 and Figure 15 that when the angle of attack gradually increases to about 50°, the estimation of the neural network method can still remain within ±1°, whereas the absolute values of the estimation error of other methods introduced in the first part exceed 1° or even more in this case [5,6,7,8,9,10,11,12,13,14,15,16,17,18,19,20,21,22,23,24,25]. This is because the combination of pressure characteristics brought about by the five pressure-measuring holes in different directions can better correspond to this specific combination of flow parameters.(2)The errors of predicting angle of attack, sideslip angle and incoming flow velocity by the neural network method will gradually increase as the angle increases (angle of attack or sideslip angle). Good prediction results can be obtained within 50°, and relatively good results can be obtained within 70°. The deviation in prediction results will increase rapidly at large angles. By referring to [26], it can be found that when the angle of attack or sideslip angle is relatively large, the flow field separation is serious, and the pressure data have relatively large fluctuations (and is unstable), which rapidly reduces the estimation accuracy at this time. When the angle is relatively small, even though there may be one or more pressure measuring hole in the airflow separation zone, the remaining pressure data can make a great contribution to the accurate estimation of parameters.(3)Due to the simple and clear relationship between the pressure and the incoming flow velocity, the error of the neural network in the estimation of the incoming flow velocity is very small (Figure 16), which is true in the whole measured velocity range.(4)The performance of the neural network trained with filtered data is better than that of the neural network trained with raw pressure data. From the comparison between Figure 9 and Figure 10, we can see that the estimation error is reduced after filtering.(5)In a small angle range, the filtered data will slightly increase the prediction errors of the angle of attack, sideslip angle and incoming flow velocity, but the errors are still within good ranges. This conclusion can be seen in Figure 14, Figure 15 and Figure 16. Theoretically speaking, the smaller the angle of attack, the smaller the corresponding estimation error. However, these images show that when the angle of attack is relatively small (that is, the middle part), the estimation errors of these parameters exceed those of the adjacent part.(6)We also found that the performance of angle-of-attack estimation and that of sideslip-angle estimation are not similar (like Figure 14 and Figure 15). This is an anomaly because we were using a five-hole detector with center-based symmetry. We speculate that this phenomenon was caused by the difference in the ranges of angle of attack and sideslip angle in the experiment, and the difference in the data distribution used in the training process, which of course needs to be verified by more experimental data in the future.

## 6. Conclusions

In this paper, the estimation of the airflow parameters of the tail-sitter VTOL aircraft needed was studied. The feasibility and scope of application for the 5-hole-probe and ANN method for estimating air flow parameters were explored. A compound airflow-parameters-estimation method using a 5-hole probe and ANN estimation algorithm was proposed to offer a solution to face the challenges of large AOA and AOS measurement or estimation for the tail-sitter UAV. As is known, there was almost no prior work on large AOA and AOS estimation for tail-sitter UAVs. Analysis results show that the proposed method has good performance.

This paper first discussed the calculation methods used in the estimation of airflow parameters of multi-hole probes in recent years; and briefly described the principles, advantages and disadvantages, and application scopes of those various methods. After comprehensive consideration, the applicable ANN method was selected to use the pressure data, and the final neural network structure was determined through a series of experiments and comparisons. Finally, for the research object of this paper, we specially selected 5-hole pressure probes and completed the required wind-tunnel experiments.

In addition, differently from article [26], we first pre-filtered the experimental data, verifying that the pre-processing has positive significance for the subsequent training. Secondly, we optimized the results of the neural network from different aspects and obtained the neural network structure which provides the most accurate parameter estimation to a certain extent. Finally, through the analysis of more abundant experimental data, we obtained more conclusions not mentioned before, including the analysis of the measurable range, the variation trend of error, the influence of data distribution and so on.

Although it is a pity that we have not completed the onboard experiment, it can be seen from the experimental results that the neural network method can provide estimation of the angle of attack with estimated error of less than 1° within the range of ±70° of the angle of attack (which can meet the requirements of most of the flight time of the tail-sitter VTOL aircraft). It can provide sideslip-angle estimation with an estimation error of less than 1.5° within the range of ±50° (which can meet the requirements in almost all cases of tail-sitter VTOL aircraft’s flight) and can provide very accurate velocity estimation when the speed is low (less than 25 m/s). All these indicate that the 5-hole probe based on the neural network can provide much better estimation accuracy than other traditional methods for the estimation of the angle of attack, sideslip angle and incoming flow velocity of the tail-sitter VTOL aircraft, which is of great significance for the subsequent development of the tail-sitter VTOL aircraft.

## Figures and Tables

**Figure 1 sensors-23-00417-f001:**
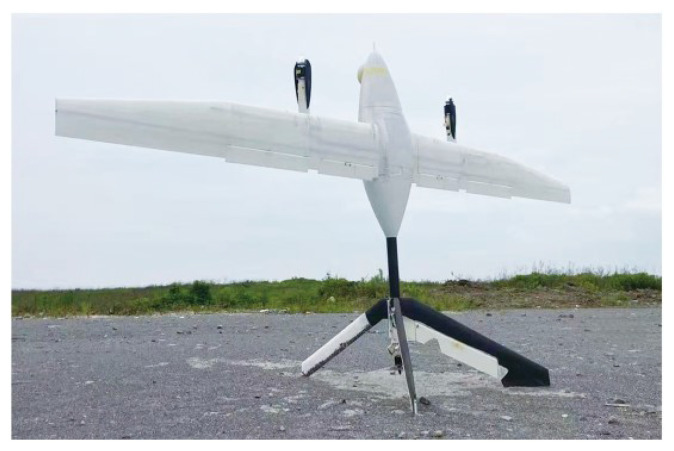
ETS-20 Tail-sitter VTOL Aircraft.

**Figure 2 sensors-23-00417-f002:**
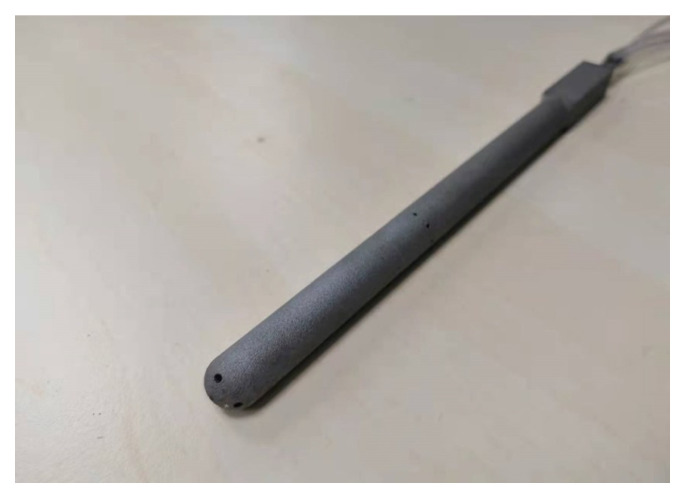
The five-hole probe.

**Figure 3 sensors-23-00417-f003:**
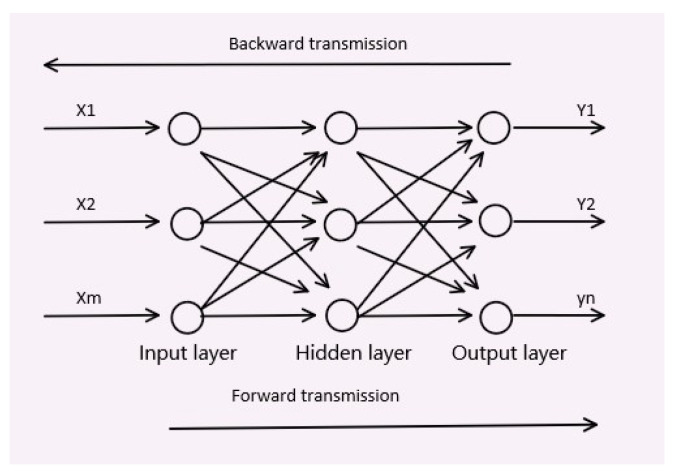
A simple BP neural network.

**Figure 4 sensors-23-00417-f004:**
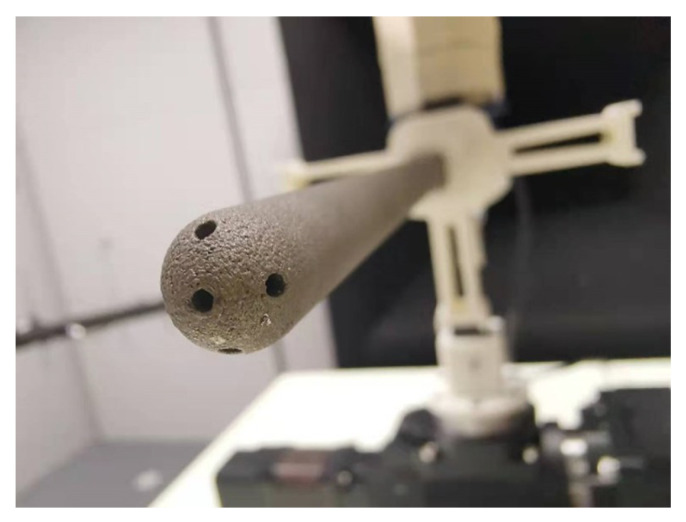
Five-hole probe made by 3D metal printing.

**Figure 5 sensors-23-00417-f005:**
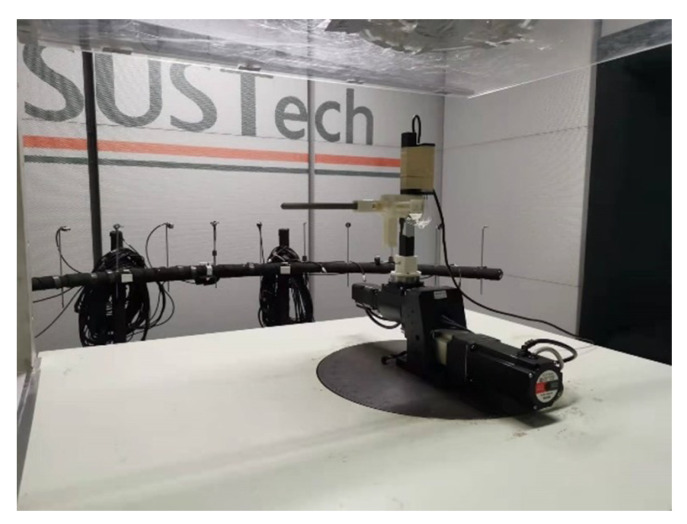
Turntable.

**Figure 6 sensors-23-00417-f006:**
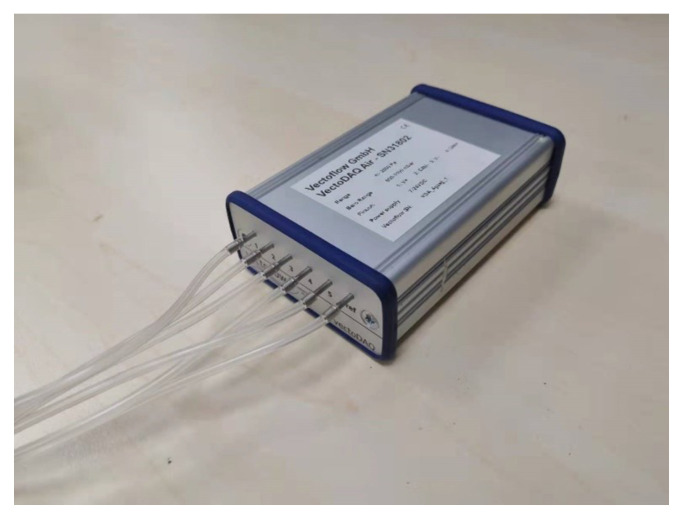
VectoDAQ Air Date Computer.

**Figure 7 sensors-23-00417-f007:**
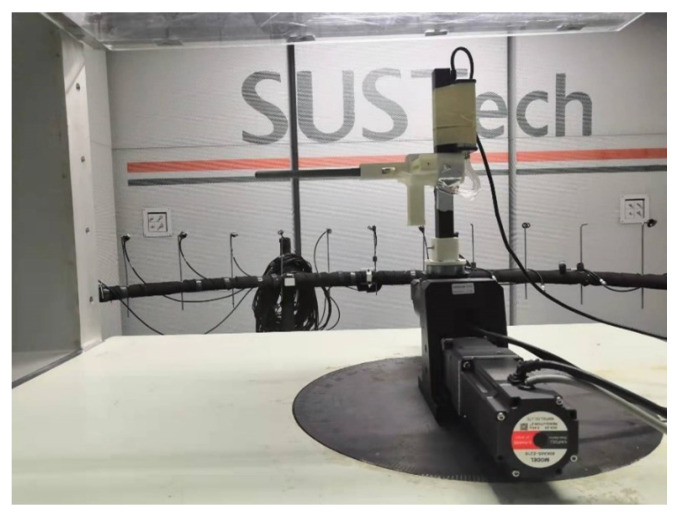
The wind-tunnel test section.

**Figure 8 sensors-23-00417-f008:**
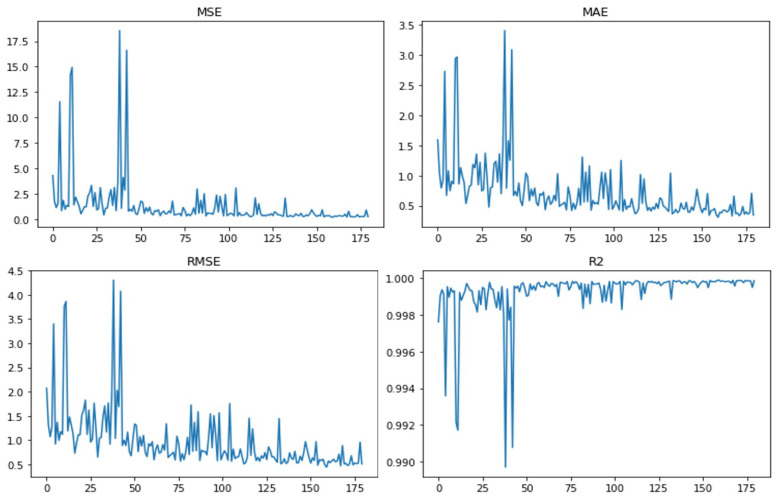
Training performance of the 180 different neural networks.

**Figure 9 sensors-23-00417-f009:**
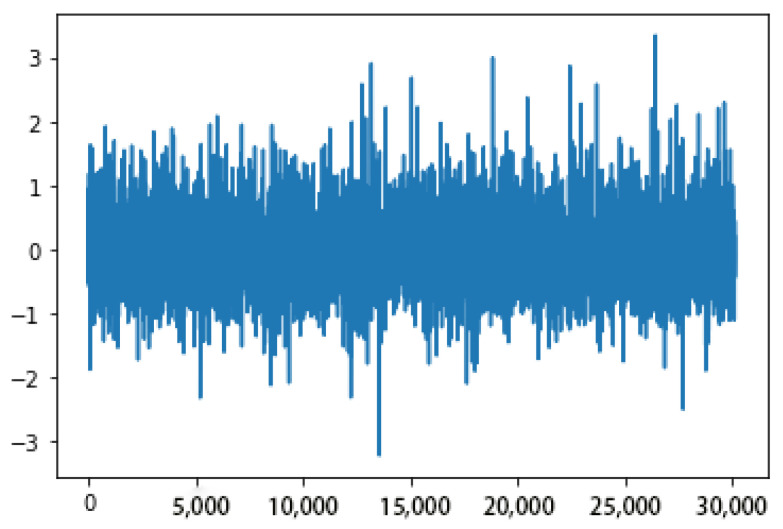
Partial estimation error before filtering.

**Figure 10 sensors-23-00417-f010:**
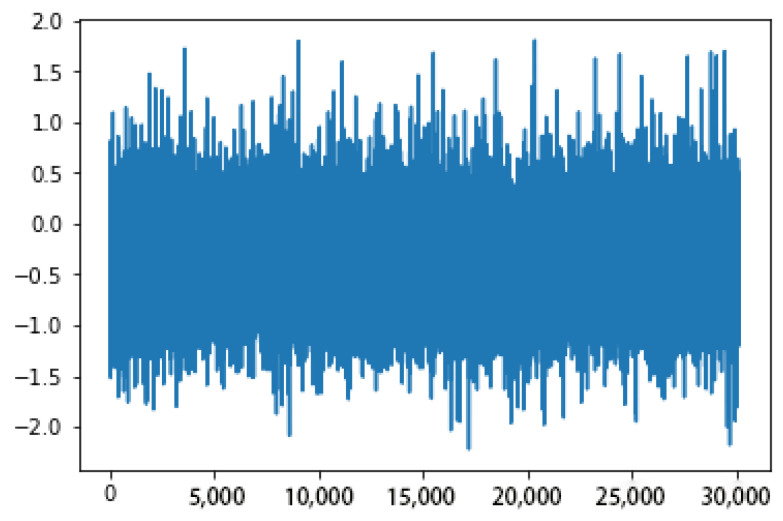
Partial estimation error after filtering.

**Figure 11 sensors-23-00417-f011:**
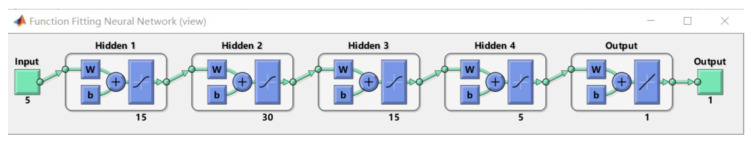
The final structure of the ANN.

**Figure 12 sensors-23-00417-f012:**
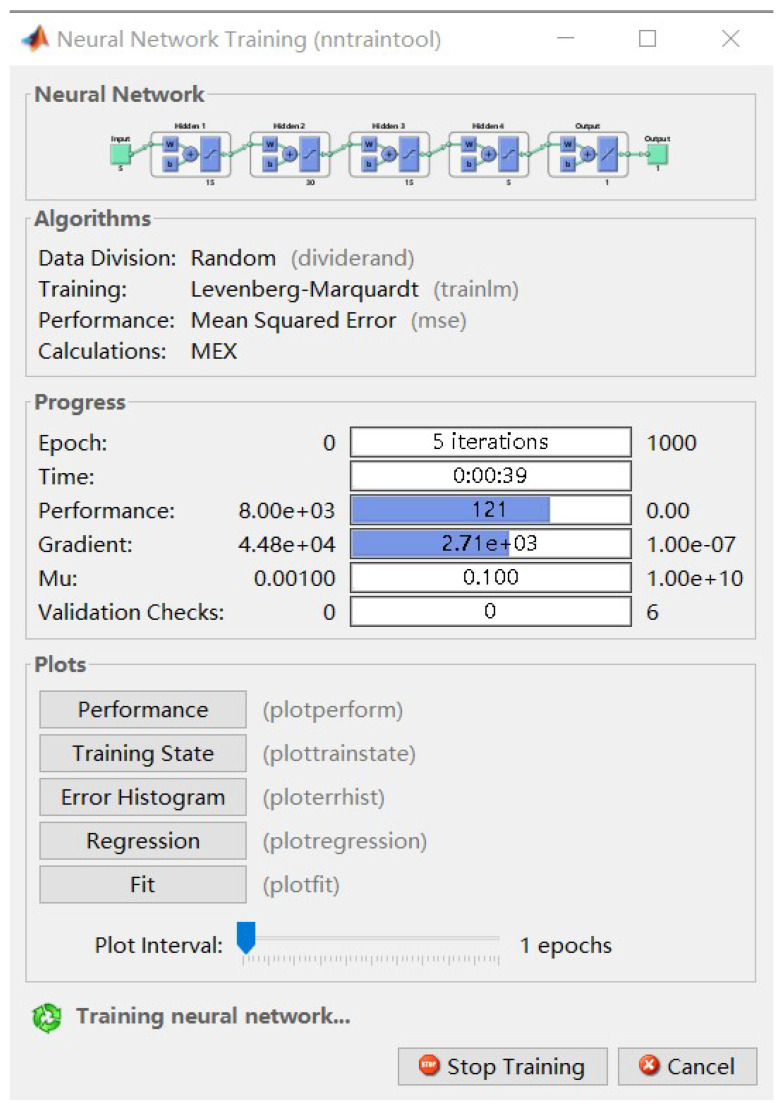
The neural network tool in Matlab.

**Figure 13 sensors-23-00417-f013:**
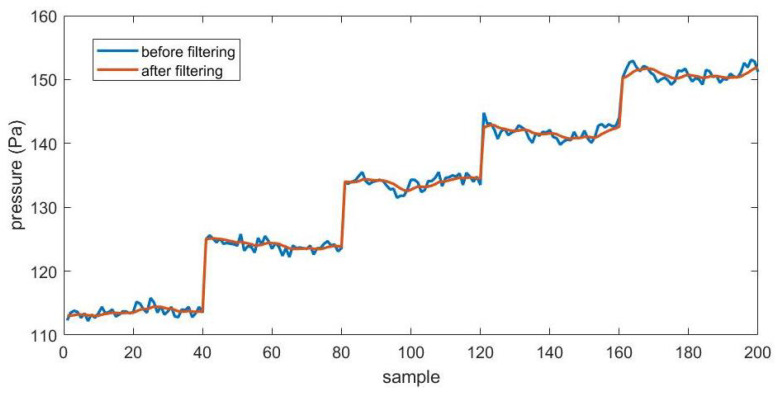
Comparison of the fluctuations of some experimental data before and after filtering.

**Figure 14 sensors-23-00417-f014:**
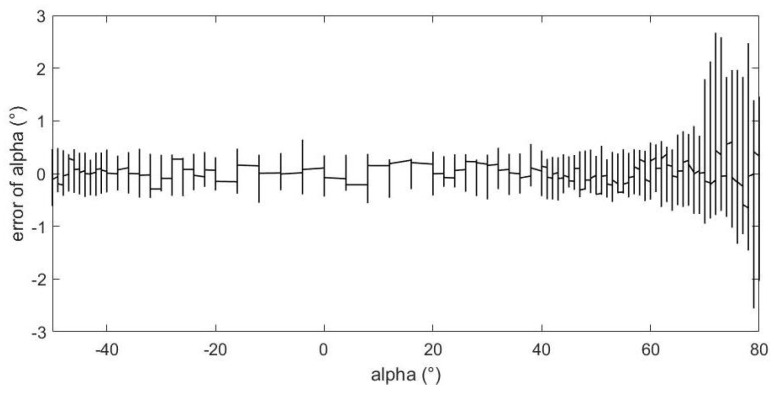
The relationship between angle of attack estimation error and angle of attack value.

**Figure 15 sensors-23-00417-f015:**
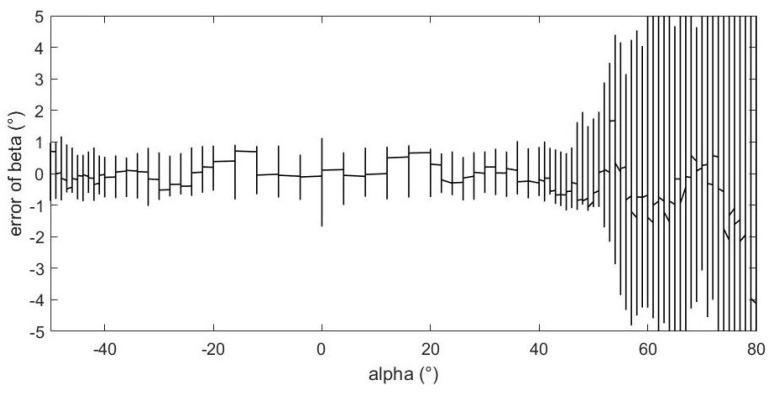
The relationship between sideslip angle estimation error and angle of attack value.

**Figure 16 sensors-23-00417-f016:**
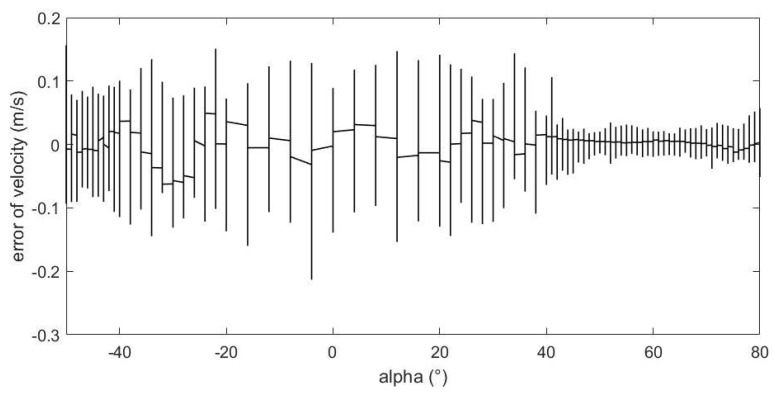
The relationship between velocity of incoming flow estimation error and AoA value.

**Figure 17 sensors-23-00417-f017:**
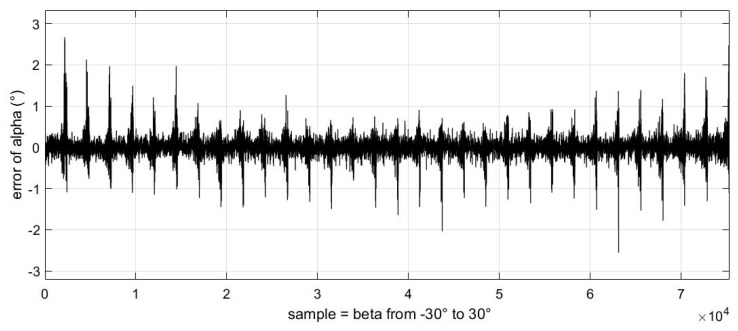
The relationship between angle of attack estimation error and sideslip angle value.

**Figure 18 sensors-23-00417-f018:**
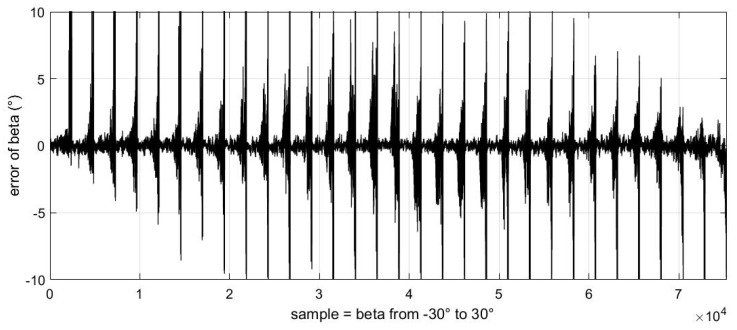
The relationship between sideslip angle estimation error and sideslip angle value.

**Figure 19 sensors-23-00417-f019:**
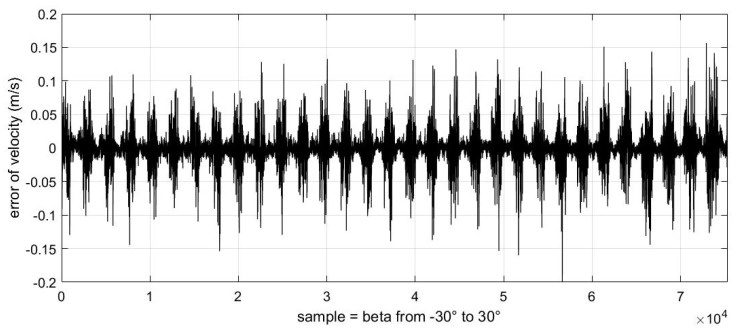
The relationship between velocity of incoming flow estimation error and sideslip angle value.

## Data Availability

Due to the large amount of data, we are sorry that we will not provide research data. Please feel free to contact the author if necessary.

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
