# Peer review of "Estimation of Airflow Parameters for Tail-Sitter UAV through a 5-Hole Probe Based on an ANN"

_sensors, 2022, doi:10.3390/s23010417_

Round 1
Reviewer 1 Report
This paper studies the estimation of the airflow parameters of the tail-sitter VTOL aircraft. I have the following concerns:
1. Please provide a contribution paragraph at the end of Section 1 (Introduction).
2. Figures should be mentioned in the text.
3. Figure 3 seems to be taken from somewhere. Please cite if it is taken.
4. From Figures 11 and 12, it seems you used MATLAB for the simulation. Please mention which software and tools are used to conduct the simulation.
5. In order to determine quantitatively the best model, statistical error measures such as the mean error (ME), the mean square error (MSE), and the mean absolute error (MAE) should be given.
6. Please proofread the manuscript. There are a lot of typos.
Reviewer 2 Report
There is a good research direction in this paper, that is, ANN's prediction of air flow is applied to fixed-wing vertical take-off and landing (VTOL) UAV。A good prediction effect can be beneficial to the operation of UAVs. There are two main points in this paper. The first is the data acquisition experiment of the angle of attack, sideslip angle and speed of incoming flow. The second are the ANN based estimation method and error analysis. But the following problems cannot be ignored:
1. High and low speed are mentioned both in the abstract and the introduction. Are there references available? What about the range of the high and low speed? Please add the description.
2. (line 29) About the traditional measurement devices, like pitot tube, vane angle sensor, are not unsuitable for large angle measurement. Whether relevant descriptions are mentioned in previous review articles?
3. (line 30) In the introduction, it is mentioned that the traditional measurement devices have negative impact on the shape design of UAV. Is there any previous research on this?
4. (line 30) The performance of the traditional measurement devices is poor at the low speed. Are there any pre-experimental comparisons or references available to draw this conclusion?
5. (line 32) More people choose to use multi-hole probe to replace traditional measurement equipment. Whether there are references to previous work.
6. In Figure 13, there is no legend to know the difference between the data before and after filtering. In addition, it is mentioned later that filtering will increase the volatility at low angles, but it cannot be clearly shown in the figure. Please modify the picture to improve the clarity. At the same time, it is suggested to increase the calculation of data volatility to enhance readability.
7. Please add the description of whether the ranges of the angle of attack, sideslip angle respectively set in the test Settings are related to the flight characteristics of fixed-wing vertical take-off and landing (VTOL) UAV in the paper.
8. Please explain why only three wind speed levels are set for the test.
9. In chapter 4. Optimization of estimation method based on ANN, the author compares the effects of neural networks with different numbers of neurons and different layers and selects the optimal network. Can you compare and express other parameters of this network with those of other networks with different structures? The following parameters, such as prediction accuracy, convergence speed and training time, will be used in the comparison of prediction models in most papers.
10. Six conclusions are put forward begin line 285 of the paper. The analysis depth of error data is not enough. For example, the first point, the neural network can well predict whether there are corresponding evaluation indicators or specific data. The fourth point mentioned that the effect of data filtering is better than the original pressure, and there is no corresponding comparative test comparison in this paper.
11. There are a large number of repeated descriptions between the experimental conclusions in line 285 and the conclusions in line 307. Is it necessary to adjust the contents?
12. The application of wind speed measurement in fixed-wing vertical take-off and landing (VTOL) UAV is proposed in the introduction part of this paper. The actual test of the device in UAV is not seen in the test part, and the impact of the prediction effect of the device on the work of UAV is not discussed in the conclusion part. Please add discussion, to enhance the integrity of this article.
13. The conclusion part is more about stating the results found in the experiment, and there is a lack of discussion and induction of the law behind the results. It is suggested to strengthen the in-depth discussion in the conclusion part.
Round 2
Reviewer 1 Report
I think the authors have addressed my comments well. I have no further comments.
Reviewer 2 Report
The author made corresponding modifications to the problems raised, and the content was improved to some extent. The literature in the introduction section was supplemented in detail. In addition, the author made a detailed and in-depth analysis in the conclusion. The significance of the AI method to estimate the angle of attack, sideslip angle and speed of incoming flow for the work of fixed-wing vertical take-off and landing (VTOL) UAV is described in detail. It is expected that the author will make further research on the practical application of the device in fixed-wing vertical take-off and landing (VTOL) UAV.